# Different Types of Cellular Stress Affect the Proteome Composition of Small Extracellular Vesicles: A Mini Review

**DOI:** 10.3390/proteomes7020023

**Published:** 2019-05-23

**Authors:** Agata Abramowicz, Piotr Widłak, Monika Pietrowska

**Affiliations:** Center for Translational Research and Molecular Biology of Cancer, Maria Sklodowska-Curie Institute—Oncology Center, Gliwice Branch, 44-101 Gliwice, Poland; agata.abramowicz@io.gliwice.pl (A.A.); piotr.widlak@io.gliwice.pl (P.W.)

**Keywords:** exosome, extracellular vesicles, genotoxic stress, heat shock, hypoxia, mass spectrometry, oxidative stress, proteome

## Abstract

Extracellular vesicles (EVs) are well-known mediators of the cellular response to different stress factors, yet the exact mechanism of their action remains unclear. Hence, the characterization of their cargo, consisting of proteins, nucleic acids, and different classes of metabolites, helps to elucidate an understanding of their function in stress-related communication. The unexpected diversity and complexity of these vesicles requires the incorporation of multiple technologically advanced approaches in EV-oriented studies. This mini review focuses on the invaluable role of proteomics, especially mass spectrometry-based tools, in the investigation of the role of small EVs in their response to stress. Though relatively few experimental works address this issue to date, the available data indicate that stress conditions would affect the composition of protein cargo of vesicles released by stressed cells, as evidenced by the functional importance of such changes in the context of the response of recipient cells.

## 1. Introduction

Extracellular vesicles (EVs) are a heterogeneous group of vesicles conventionally assigned to three classes based on their cellular origin: Apoptotic bodies, microvesicles, and exosomes [1]. Though this classification has been well-established in the scientific community, the currently available and commonly used techniques of the isolation of EVs do not provide efficient separation of individual subclasses [2,3]. Therefore “simplified” and less specific nomenclature is preferred nowadays, like small EVs (i.e., <200 nm) and medium/large EVs (>200 nm) [3,4]. The class of small EVs is enriched in exosomes that are vesicles of endosomal origin with diameter up to 150 nm but also includes other EVs (e.g., small microvesicles formed by budding of the plasma membrane). Medium/large EVs consists mainly of microvesicles with diameter up to 500 nm, however, especially under stress conditions, apoptotic bodies are also included in this group [2,5]. Independently on their origin, EVs have been recognized as carriers of broad range of particles like proteins, lipids and nucleic acids that are probably strictly related to function of individual types of EVs [6,7].

The sudden interest of EVs resulted from the discovery of their key role in intercellular communication, ability to modulate the immune response, and involvement in the development of cancer and neurodegenerative diseases [8]. From the point of view of translational research, particularly noteworthy are studies concerning EVs-mediated signaling in the response to cellular stress that is crucial to understand the mechanisms of disease development as well as the effectiveness of therapeutic interventions. Proteomic approaches are beneficial for EVs-oriented studies because they provide new knowledge on the cargo of vesicles, which helps in understanding their functional roles. Since knowledge of a role of EVs in the immune system regulation and response to stress caused by pathogens has been presented very recently in a few excellent reviews [9,10,11], this mini review focuses on current knowledge about the proteome of small extracellular vesicles released by cells subjected to environmental physicochemical stress caused by heat, DNA damaging factors, oxidation, and oxygen or nutrient deficiency. 

## 2. Heat Shock

Heat is one of the most commonly occurring stress factors that require the development of specific cytoprotective mechanisms. Elevated temperature causes protein denaturation, unfolding, and aggregation that further lead to inactivation and loss of functionality. Hence, proteins are among the molecules of which the negative effects of hyperthermia are the most visible, and heat shock is an archetypical proteo-toxic type of cellular stress. The improper function of proteins disturbs cellular homeostasis, and forces the activation of the heat shock response (HSR), of which the main purpose is to limit the damage of crucial cellular components as well as stopping processes associated with proliferation and growth until elimination of the stress factor and the repair of damage [12,13]. The major effectors of the HSR are heat shock proteins (HSPs). These are molecular chaperones responsible for protein folding, unfolding, and/or refolding in either normal or stressed conditions [14]. Several HSP70/HSPA proteins (HSPA1A, HSPA5, HSPA8) and HSP90/HSPC proteins (HSP90AA1, HSP90AB1) are constitutive components of EVs regardless of the stress conditions [15,16]. HSP proteins are the most frequently reported exosomal proteins present on the list provided by ExoCarta, EVpedia, and Vesiclepedia databases [17,18,19]. Studies performed using peripheral blood mononuclear cells revealed that the level of stress-inducible HSPA1 in EVs increased after hyperthermia (40 or 43 °C for 1 h) without a significant change in the vesicle number [20]. On the other hand, an only small increase in the number of heat shock proteins (HSPB1, HSPA8, HSPA1, and HSPC1) was observed in EVs released by B-lymphoblastoid cells subjected to hyperthermia [21]. Hyperthermia increased the level of heat shock proteins (HSPA1 and HSPD1) in EVs from the ascitic fluid of gastric cancer patients. Such EVs promoted the maturation of dendritic cells and increased anti-tumor activity of cytotoxic T lymphocytes [22], which could reflect a role of HSPs in immune-related functions (e.g., antigen presentation) [23]. Moreover, EVs released by heat-stressed cells were postulated to mediate the bystander effect in unstressed cells. EVs released from MCF7 cells subjected to hyperthermia (45 °C for 1 h) reduced the viability and increased the level of DNA damage and apoptosis in recipient cells when compared to EVs released by control cells. On the other hand, vesicles from heat-shocked cells reduced the sensitivity of recipient cells to subsequent thermal stress, indicating their role in the adaptation to hyperthermia [24]. Surprisingly, despite promising results suggesting the significant role of EVs in the HSR, there is an evident lack of comprehensive proteomic studies showing the whole range of changes in protein composition of EVs released by cells subjected to hyperthermia. 

## 3. Genotoxic Stress

The protection of genetic material is critical to maintain the cellular functions of an organism at all levels of its organization. Therefore, DNA damage is particularly lethal and requires complicated molecular machinery responsible for maintaining genomic stability through DNA repair. A multitude of crucial regulated aspects include cell cycle checkpoints, DNA repair, and the eventual decision whether to activate programmed cell death in cells with unrepairable DNA damage [25]. Genotoxic stress is caused by a plethora of factors that damage DNA. These include ionizing and ultraviolet radiation, alkylating agents, reactive oxygen species, and inhibitors of topoisomerases and helicases [25]. It is well-known that genotoxic stress affects the release and composition of extracellular vesicles [26,27]. Ionizing radiation (IR) is a type of genotoxic agent that is best studied in the context of EVs-mediated communication. It has been postulated that small EVs released by irradiated cells are involved in different aspects of the systemic response to radiation, including, but not limited to, the radiation-induced bystander effect (RIBE) [28,29,30,31]. Proteomic profiling of EVs released from FaDu cells (derived from a hypopharynx carcinoma) irradiated with a single 2 Gy IR dose revealed increased levels of proteins involved in transcription and translation, chaperones, ubiquitination-related factors, and proteasome components [32]. A similar profile of the proteome of the small EVs released by UM-SCC6 cells (another cell line derived from head and neck cancer) irradiated with three different doses (2, 4, and 8 Gy) allowed for the identification of over 1200 exosomal proteins including about 470 proteins specifically affected by irradiation. Among overrepresented pathways associated with IR-affected proteins were those involved in the DNA damage response and repair of double strand breaks (e.g., XRCC5 or XRCC6) [33]. This observation was consistent with other reports showing that small EVs released by irradiated cells promotes the survival of recipient cells. Mutschelknaus et al. demonstrated that the radioprotective effect of EVs released by irradiated cells derived from head and neck cancer (FaDu and BHY cell lines) was achieved by stimulation of DNA repair in recipient cells [31]. Proteomic data presented by the same group indicated that EVs released by irradiated cells may also promote migration of recipient cells [34]. An analysis of EVs released by BHY cells irradiated with 6 Gy dose revealed 39 up-regulated and 36 down-regulated proteins and an overrepresentation of processes associated with regulation of cell motility when compared with EVs released from non-irradiated cells [34]. More recently, transcriptome and proteome profiling of EVs released by glioma cells irradiated with 3 and 12 Gy was reported [35]. The EVs from irradiated cells enhanced the proliferation and radio-resistance of recipient cells in vitro, as well as promoted an increased tumor burden in vivo. Proteomic profiling revealed that, among approximately 1000 proteins detected in analyzed EVs, there were approximately 300 proteins present only after irradiation. These proteins were associated with pathways that either directly or indirectly were involved in the cellular response to ionizing radiation like proteasome pathway, the Notch signaling pathway, the Jak-STAT signaling pathways, and cell cycle regulation [35]. Hence, the results of available proteomic studies support the idea that EVs released by irradiated cells mediate pro-survival signals that increase radio-resistance of recipient cells. Interestingly, this observation is in contrast to the other hypothetical role of EVs released by irradiated cells, which would be the mediation of cytotoxic and genotoxic effects of RIBE [28]. Experiments with whole blood samples clearly indicated capabilities of EVs released by irradiated blood cells to modulate immune system. This ex vivo irradiation stimulated secretion of EVs enriched in proteins involved in acute phase response signaling, complement system, LXR/RXR and FXR/RXR activation, IL-12 signaling and regulation of macrophage functions [36]. Fragmentary data concerning the proteome composition of EVs released by cells treated with other types of genotoxic factors are also available. One mass spectrometry (MS)-based study revealed fibronectin as a crucial pro-survival component of EVs released from melanocytes in response to UV-B radiation [37]. Another MS-based study of EVs released by glioblastoma stem-like cells (GSC4) in response to temozolomide treatment revealed a group of proteins involved in the modulation of cell adhesion [38]. It is well-documented that EVs are involved in the mediation of resistance to cisplatin [27,39]. Hence, it is noteworthy that analysis of EV phosphoproteome (by the Proteome Profiler Human Phospho-MAPK Array Kit) revealed that among the phosphoproteins up-regulated by this treatment were JNK2, JNK-pan, p38α, and p53. Moreover, p38 and JNK proteins were considered as the most likely culprits in observed chemoresistance and increased invasiveness of recipient cells [40]. 

## 4. Oxidative Stress

Though reactive oxygen species (ROS) are a byproduct and a metabolite of normal physiological metabolic processes, they can be dangerous if not neutralized properly. ROS can react with all crucial cellular molecules including lipids, proteins, and nucleic acids, generating damaged, non-functional products. Sometimes cell protection mechanisms are inefficient or there is a rapid increase in levels of free radicals produced by external sources like IR. Then the imbalance between the production of ROS and antioxidants leads to cellular oxidative stress [41]. The influence of free radicals on the cell can be different depending on the dose and type of free radical species. Low doses of ROS can stimulate cell proliferation, while higher doses can result in cell cycle arrest or even apoptotic, senescent, or necrotic cell death [42]. The mechanism of cellular response to reactive oxygen species are under intensive investigation, which provides interesting information about the role of EVs in this process. The cytoprotective properties of EVs were demonstrated in studies of EVs released from mouse mast cell line MC/9 exposed to H_2_O_2_ [43]. In this model, pre-incubation of cells with EVs derived from ROS-exposed cells increased the tolerance of recipient cells to subsequent exposure to hydrogen peroxide, as evidenced by their higher viability compared to cells incubated with EVs derived from non-stressed cells. A similar effect was observed in bovine granulosa cells exposed to oxidative stress [44]. EVs derived from stressed cells were significantly enriched with mRNA of the *NRF2* gene, one of the crucial regulators of cellular resistance to oxidants [45]. Moreover, the elevated level of transcripts of *NRF2* and *NRF2*-related genes was also detectable in recipient cells after co-incubation with stress-related EVs [44]. Available proteomic studies support the hypothesis of a significant role for small EVs in response to oxidative stress by showing the significant influence of free radicals on protein cargo of EVs. Proteomic profiling of EVs released by primary amnion epithelial cells (AEC) in response to oxidative stress induced by cigarette smoke extract allowed for the identification of 48 unique proteins in stress-related EVs when compared to EVs released in control conditions. Vesicles derived from stressed cells were enriched in proteins related to ERK/MAPK, PI3K/AKT, epithelial adherens junction signaling pathways, and eosinophilic inflammation disease, whereas underrepresented proteins were involved in LPS/IL-1 mediated inhibition of RXR. Moreover, Western blot analysis demonstrated a higher level of active phosphorylated p38 MAPK in the EVs from stressed cells. This indicated that EVs could reflect the physiologic status of the parental cells, and they might be used to monitor the risk for pregnancy [46]. A clinically relevant study of EVs in the context of age-related macular degeneration (AMD) was also conducted [47]. The retina is characterized by intensive oxygen consumption due to the high metabolic rate of this tissue, and it is prone to pathological changes related to exposure of high levels of ROS. Hence, oxidative stress is one of the main factors involved in retinal age-related macular degeneration. To test the hypothesis that specific phosphoproteins and signaling factors identified in the vitreous body of patients with AMD can be delivered by EVs, proteomic profiling of EVs released by human retinal pigment epithelium cells (ARPE-19) under oxidative stress conditions were performed. The reverse phase protein microarray (RPMA) analysis allowed for the identification of 72 proteins carried by EVs, including 41 post-translational modifications (e.g., phosphorylations). Some of the phosphoproteins identified in EVs from cell line overlapped with those identified in samples of patients with AMD, including PDGFRβ, VEGFR2, and c-kit [47,48]. Among phosphoproteins affected in small EVs in response to oxidative stress, there were those related to cell death (Smac/Diablo, Bak, Apaf-1), cell proliferation, survival and growth (Akt, PDK1, and ERK1/2), and metabolism (LDHA, Acetyl-CoA Carboxylase) [47]. This finding indicates the potential diagnostic role of EV proteins in oxidative stress-related pathologies of the retina.

## 5. Hypoxic Stress

Hypoxia is a state of deficiency of oxygen supply. As O_2_ is crucial for cellular bioenergetics, its deficiency is a powerful stress factor. To manage this condition, the cell arrests the cell cycle and reduces energy consumption [49]. Characteristic pathways activated in response to hypoxic stress are related to mTOR kinase and hypoxia-inducible factors (HIFs) [49]. However, recent studies clearly demonstrate that the response to oxygen deficiency goes well beyond the cellular level, and EVs are postulated to be key players in cell-to-cell communication under hypoxic conditions. In vitro studies on three breast cancer cell lines: MCF7, SKBR3, and MDA-MB231 revealed a significant increase of EVs level in culture media when cells were cultured at 1% and 0.1% O_2_ tension, with a stronger effect for lower oxygen content [50]. Small EVs secreted by glioma cells under hypoxia conditions were able to induce microvascular sprouting and promoted proliferation and survival in oxygen deficiency of HUVEC and HBMEC cells [51]. Hypoxic EVs stimulated tumor growth in the glioblastoma (GMB) xenograft mouse model. Moreover, treatment with hypoxic EVs increased tumor vascularization and decreased areas of hypoxia [51]. These findings have been supported by later results obtained by Kore et al. [52], who showed a significant increase in the level of proteins involved in angiogenesis and extracellular matrix remodeling, including TSP-1, LOX, VEGF, and ADAMTS1, in EVs released by hypoxic U87MG glioblastoma cells. Furthermore, treatment of endothelial progenitor cells with hypoxic EVs stimulated tube forming ability when compared with normoxic EVs released by U87MG cells [52]. Moreover, proteomic profiles of EVs released in response to hypoxic stress differ from those released under normal conditions also in other types of cells, confirming that the EV-related pathway is a part of the general mechanism of response to oxygen deprivation. Proteomic analysis of EVs released in vitro by mice cardiac fibroblasts cultured for 24 h in 21% O_2_ (normoxia) and 2% O_2_ (hypoxia) allowed for the identification of 1,752 and 1,616 proteins, respectively [53]. Quantitative analysis revealed significant differences in expression of 144 proteins (71 proteins were up-regulated and 73 proteins were down-regulated) in EVs released under hypoxic conditions. Proteins enriched in EVs released under oxygen deficiency were mainly related to the extracellular compartment and mitochondria, which suggested their role in the maintenance of mitochondria damaged by O_2_ deficiency [53]. Quantitative iTRAQ-based proteomic studies on EVs released by endothelial cells confirmed the significant influence of hypoxia on EV content [54]. A total number of 1,354 proteins were identified in EVs released by cells cultured under normoxia and hypoxia (2% O_2_). Similar to the fibroblast-based study, among the most hypoxia-up-regulated EV proteins were those connected with the extracellular compartment, which supported the potential role of EVs in extracellular matrix remodeling under hypoxic conditions. However, proteomic studies performed on EVs derived from hypoxia-stressed (1% O_2_) prostate cancer cells did not reveal enrichment of extracellular matrix proteins [55]. Nevertheless, data presented in this report [55] indicated that proteins present in hypoxia-induced EVs may be involved in remodeling of epithelial adherens junction, actin-related cytoskeletal rearrangements, and regulation of endocytosis.

## 6. Nutrient Stress

The availability of nutrients is crucial for metabolic homeostasis and for the proper function of the cell. One of the most important nutrients is glucose, and both deprivation and excess of glucose can cause cellular stress [56,57]. Recent studies indicate that EVs may also play a role in glucose-related disorders. It was shown that glucose starvation increases the secretion of EVs from the rat myoblast H9C2 cell line, as well as from rat primary cultured cardiomyocytes [58]. Moreover, glucose starvation affects the protein cargo of released EVs. Vesicles released under starvation conditions contained primarily proteins associated with metabolism (small molecule catabolic processes, glycolytic process, or pyruvate metabolic process) and vesicle trafficking, while vesicles released under normal availability of glucose contained proteins involved in cell growth and adhesion. In functional studies, EVs released from starved cells had a stronger pro-proliferative and pro-angiogenic effect on endothelial cells than EVs released in control conditions [58]. Furthermore, vesicles released from rat glucose-starved cardiomyocytes deliver to endothelial cells functional glucose transporters and glycolytic enzymes and are able to stimulate glucose uptake and pyruvate synthesis in recipient cells [59]. This confirmed the active participation of small EVs in the mechanism of response to nutrient starvation and glucose homeostasis.

## 7. Conclusions

Available data, though still far incomplete, clearly demonstrate that extracellular vesicles play an important role in the cellular response to stress. The proteome composition is markedly affected in vesicles released by cells exposed to cellular stress; representative studies that focused on the impact of different stress conditions on the proteome of EVs are listed in Table 1. However, no universal signature of stress could be identified in EV proteome at present. Moreover, different protein sets were detected in EVs released by cells treated with similar stress conditions. This discrepancy apparently reflected different biological models (cell types and stress dosages) and specific types of EVs analyzed. Nonetheless, in spite of different subsets of EV proteins enriched in different models, there were generally similar biological functions of such proteins that could be actually or potentially imposed on recipient cells. In the majority of cases, proteins enriched in EVs released under stress conditions could be involved in regulation of proliferation and cell survival, reshaping of the extracellular environment, migration as well as angiogenesis (Figure 1). Further studies are necessary to develop a more detailed understanding of the role of EV-mediated stress-induced mechanisms. Importantly, however, the application of proteomic approaches has already proven their accuracy and value towards extending knowledge on the EV-mediated mechanisms of inter-cellular communication involved in response to stress conditions.

## Figures and Tables

**Figure 1 proteomes-07-00023-f001:**
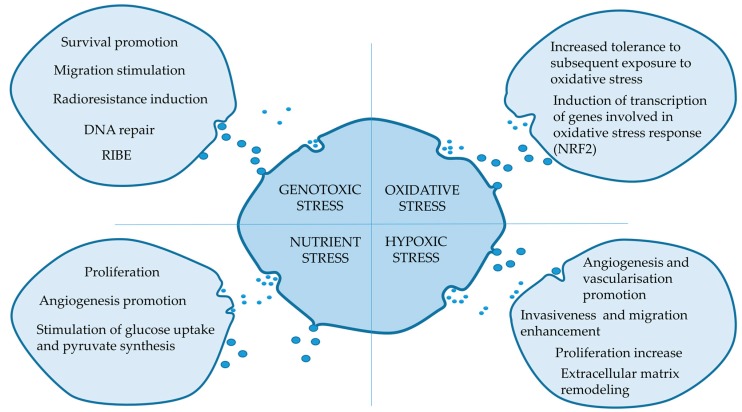
Potential impact of extracellular vesicles (EVs) released by stressed cells in recipient cells.

**Table 1 proteomes-07-00023-t001:** Examples of studies on the impact of different stress conditions on the proteome composition of EVs released by stressed cells.

Stress Factor	Research Model	Method of EV Purification	Proteomics Approach	Major Observation (Pathways Associated with Affected Proteins)	Ref.
Ionizing radation (2-8 Gy)	UM-SCC6 head and neck cancer cells, 24 h	SEC	LC-MS/MS	Proteins involved in DNA metabolic processes and DNA repair	[33]
Ionizing radation (6 Gy)	BHY head and neck cancer cells, 24 h	UC	LC-MS/MS	Up-regulated proteins: FGFR1, HSP90AA1, HSP90AB1, HSP90B1, and VTN	[34]
Ionizing radiation (3, 12 Gy)	U87 glioma cells, 48 h	Precipitation	LC-MS/MS	Affected pathways: proteasome, Jak-STAT signaling, cell cycle, and Notch signaling	[35]
Oxidative stress (CSE)	AEC amniotic epithelial cells, 48 h	UC	LC-MS/MS	Affected pathways: ERK/MAPK, epithelial adherens junctions, and PI3K/AKT signaling	[46]
Oxidative stress (MV)	ARPE-19 retina epithelium cells, 24 h	UC	RPMA	Down-regulation of phosphoproteins involved in cell survival and proliferation, and up-regulation of phosphoproteins involved in cell death and metabolism	[47]
Hypoxia (2% O_2_)	HMEC-1 endothelial cells, 24 h	UC	LC-MS/MS	Affected pathways: extracellular matrix (ECM) rearrangements	[54]
Hypoxia (2% O_2_)	Mouse cardiac fibroblasts, 24 h	UC	LC-MS/MS	Affected pathways: ECM rearrangements and mitochondria maintenance	[53]
Glucose starvation	Rat neonatal cardiomiocytes, 48 h	UC	LC-MS/MS	Affected pathways: protein transport and metabolism	[58]

Showed are types of cells producing EVs to culture media and times of EV collection after/during stress factors. Methods of EV purification: SEC—size exclusion chromatography, UC—ultrafiltration; oxidative stress-inducing factors: CSE—cigarette smoke extract, MV—methyl viologen; RPMA—reverse phase protein arrays.

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
