# Peer review of "Different Types of Cellular Stress Affect the Proteome Composition of Small Extracellular Vesicles: A Mini Review"

_proteomes, 2019, doi:10.3390/proteomes7020023_

Round 1
Reviewer 1 Report
Dear editor and authors,
In the review article „Different type of extracellular stress affects the proteome composition of extracellular vesicles: a mini-review” the authors summarize the current knowledge about stress-induced proteome changes in extracellular vesicles. In general I consider this manuscript as a valuable approach to briefly update the scientific community. However, I would appreciate an attempt to summarize and evaluate the data.
I suggest to extend the conclusion section by a comparison of the discussed data. Are there common stress responsive protein or pathways which are generally changed in response to stress. Findings can be illustrated in a figure.
Why are these five types of stress inducers selected? Concerning the important role of extracellular vesicles in the immune system I miss the discussion of proteome changes after infections or in response to host-pathogen interactions. There are several interesting publications available (e.g. Cyprik et al., 2017; Xiaofang Jia et al., 2017; Jeannin et al., 2018)
In paragraph 3 (genotoxic stress) only proteome changes in tumor cells are discussed. I think it is important to mention that also normal tissues respond to genotoxic stress (e.g. Gum-Ho Bin et al., 2016, melanocyten after UV or Yentrapalli et al, 2017, whole blood after ionizing radiation)
Extracellular vesicles are a rapidly evolving area, therefore I suggest to replace some articles in the introduction section by more recent publications.
Title should be: Different typesof extracellular stress affects the proteome composition of extracellular vesicles: a mini-review
Author Response
Response to Reviewer #1
Q1: I suggest to extend the conclusion section by a comparison of the discussed data. Are there common stress responsive protein or pathways which are generally changed in response to stress. Findings can be illustrated in a figure.
A1: Thank you for your valuable suggestion. Currently, no universal signature of stress could be identified in EV proteome and different protein sets were detected in EVs released by cells treated with similar stress condition (this discrepancy apparently reflected different biological models and specific types of EVs analyzed). Nevertheless, in spite of different subsets of EV proteins enriched in different models, there were generally similar biological functions of such proteins that could be actually or potentially imposed on recipient cells. In the majority of cases proteins enriched in EVs released under stress conditions could be involved in regulation of proliferation and cell survival, reshaping of the extracellular environment and migration as well as angiogenesis. A graphic with summary of the most commonly stress-related pathways/processes activated in recipient cells that could be presumed based on proteome composition of EVs has been added as Figure 1.
Q2: Why are these five types of stress inducers selected? Concerning the important role of extracellular vesicles in the immune system I miss the discussion of proteome changes after infections or in response to host-pathogen interactions. There are several interesting publications available (e.g. Cyprik et al., 2017; Xiaofang Jia et al., 2017; Jeannin et al., 2018).
A2. We apparently agree that cellular response to stress induced by pathogen infection (and regulation of immune functions in general) is a very interesting subject in the context of exosomes/EVs. However, a few comprehensive reviews in this field have been published very recently (Jones et al., 2018; Wang et al., 2018; Zhang et al., 2018; below). Therefore, to avoid overlapping with these papers we decided to focus on the non-biological stress factors that seems under-researched in the context of EV-mediated and in our opinion could be interesting for the scientific community.
Jones LB, Bell CR, Bibb KE, Gu L, Coats MT, Matthews QL. Pathogens and Their Effect on Exosome Biogenesis and Composition. Biomedicines. 2018;6(3).
Wang J, Yao Y, Chen X, Wu J, Gu T, Tang X. Host derived exosomes-pathogens interactions: Potential functions of exosomes in pathogen infection. Biomed Pharmacother. 2018;108:1451-1459.
Zhang W, Jiang X, Bao J, Wang Y, Liu H, Tang L. Exosomes in Pathogen Infections: A Bridge to Deliver Molecules and Link Functions. Front Immunol. 2018;9:90.
Q3. In paragraph 3 (genotoxic stress) only proteome changes in tumor cells are discussed. I think it is important to mention that also normal tissues respond to genotoxic stress (e.g. Gum-Ho Bin et al., 2016, melanocyten after UV or Yentrapalli et al, 2017, whole blood after ionizing radiation)
A3. We thank for this comment. Suggested publications have been included in the revised manuscript.
Q4. Extracellular vesicles are a rapidly evolving area, therefore I suggest to replace some articles in the introduction section by more recent publications.
A4. We thank for this comment. A few more recent articles (2018-2019) have been quoted in the revised manuscript.
Q5. Title should be: Different types of extracellular stress affects the proteome composition of extracellular vesicles: a mini-review
A5. We thank for this comment. The title has been modified accordingly.
Reviewer 2 Report
The paper by Abramowicz et al reviewed the current literature on extracellular vesicles by an original perspective, by some points are confused and need to be fixed, because many evidences have to be examined and reported more critically, trying to summarize the conclusion in a clearer manner, not only listing them. First, the paper focuses the interest on exosomes. However, based on recently established guidelines and publications (Thery et al., Journal of Extracellular Vesicles, 2018), from a practical point of view, it is very difficult to separate exosomes originating from endosomal system from microvesicles, originating from plasma membrane, because small microvesicles can share similar size and density with exosomes (Kowal et al., PNAS 2016). Therefore, the operative definition of small extracellular vesicles (enriched in exosomes) and medium/large extracellular vesicles (enriched in microvesicles) should be preferred. The term “exosomes” can be used in the text, mostly because in many of the cited paper it was used, but its current limitations must be clearly explained at the beginning of the review, because these methodological limitations can be at the bases of some contradictory results observed and reported in the review. Besides, during stress, cells tend to release many types of vesicles and this makes more difficult to separate exosomes from microvesicles with respect to unstressed conditions. As for example, cells irradiated often undergo significant level of apoptosis, so it must be explained clearly at what time points after irradiation “exosomes” were collected and the proteome examined. Some functions may be mediated by vesicles other than exosomes released upon stress. A critical conclusion on common proteins found in exosomes released upon stress condition would help to catch the overall message.
Minor points
In the Introduction authors wrote that “Their classification depends on various criteria, such as origin, size, or cargo, however, due to the dynamic development of this field of science the boundaries between these categories are still evolving [2,3].”. Although the classification depends on various factors, microvesicles and exosomes are terms that refers to a different cellular origin, and this should be clearly explained in a review.
The sentence “The sudden interest of EVs, especially exosomes, resulted from the discovery of their key role in intercellular communication” is misleading, as it suggests that only exosomes and not microvesicles and/or apoptotic bodies have a role in intercellular communication, but this is not correct, so the sentence should be rewritten
In section 2, authors wrote that “Such exosomes promoted the maturation of dendritic cells and increased anti-tumor activity of cytotoxic T lymphocytes [15].” Could author suggest a general mechanism for the immunomodulatory function of exosomes released upon heat-shock?
A graphical representation of proteins commonly found in small EVs upon stress induction would help.
Author Response
Response to Reviewer #2
Q1.(…) First, the paper focuses the interest on exosomes. However, based on recently established guidelines and publications (Thery et al., Journal of Extracellular Vesicles, 2018), from a practical point of view, it is very difficult to separate exosomes originating from endosomal system from microvesicles, originating from plasma membrane, because small microvesicles can share similar size and density with exosomes (Kowal et al., PNAS 2016). Therefore, the operative definition of small extracellular vesicles (enriched in exosomes) and medium/large extracellular vesicles (enriched in microvesicles) should be preferred. The term “exosomes” can be used in the text, mostly because in many of the cited paper it was used, but its current limitations must be clearly explained at the beginning of the review, because these methodological limitations can be at the bases of some contradictory results observed and reported in the review. Besides, during stress, cells tend to release many types of vesicles and this makes more difficult to separate exosomes from microvesicles with respect to unstressed conditions. As for example, cells irradiated often undergo significant level of apoptosis, so it must be explained clearly at what time points after irradiation “exosomes” were collected and the proteome examined. Some functions may be mediated by vesicles other than exosomes released upon stress. A critical conclusion on common proteins found in exosomes released upon stress condition would help to catch the overall message.
A1. We apparently agree with this valuable comment. Therefore the whole first paragraph of introduction has been rewritten to explain this point. Moreover, the term “exosomes” has been replaced with “small extracellular vesicles” or “EVs” in the revised manuscript. We also agree that time is an important factor in stress-oriented studies. Therefore, the information on time of EV collection was reported in the new Table 1 that summarizes relevant studies.
Q2. In the Introduction authors wrote that “Their classification depends on various criteria, such as origin, size, or cargo, however, due to the dynamic development of this field of science the boundaries between these categories are still evolving [2,3].”. Although the classification depends on various factors, microvesicles and exosomes are terms that refers to a different cellular origin, and this should be clearly explained in a review.
A2. The introduction has been rewritten in the revised manuscript (see the response to Q1).
Q3. The sentence “The sudden interest of EVs, especially exosomes, resulted from the discovery of their key role in intercellular communication” is misleading, as it suggests that only exosomes and not microvesicles and/or apoptotic bodies have a role in intercellular communication, but this is not correct, so the sentence should be rewritten.
A3. It was not our intention to suggest that only exosomes participate in intercellular communication. Therefore the phrase “especially exosomes” was removed in the revised manuscript.
Q4. In section 2, authors wrote that “Such exosomes promoted the maturation of dendritic cells and increased anti-tumor activity of cytotoxic T lymphocytes [15].” Could author suggest a general mechanism for the immunomodulatory function of exosomes released upon heat-shock?
A5. HSP are known factors involved in immune modulation via regulation of antigen presenting cells (e.g. dendritic cells) and facilitation of presentation of chaperoned peptides via MHC class I and II molecules. Therefore, increased presence of HSPs in EVs released by heat-shocked cells could contribute to observed EV-related phenomenon. This potential mechanism was mentioned in the revised manuscript.
Q5. A graphical representation of proteins commonly found in small EVs upon stress induction would help.
A5. Currently, no universal protein signature of stress could be identified and different protein sets were detected in EVs released by cells treated with similar stress condition (this discrepancy apparently reflected different biological models and specific types of EVs analyzed). Nevertheless, in spite of different subsets of EV proteins enriched in different models, there were generally similar biological functions of such proteins that could be actually or potentially imposed on recipient cells. Hence, a graphic that illustrates pathways/processes related to proteins enhanced in EVs released by stressed cell has been included as a new Figure 1.
Reviewer 3 Report
The mini-review from Abramowicz et al., is informative and well-written. It is appropriate for this Special Issue in Proteomes. I have minor editorial comments below. In addition, although not necessary, the mini-review would benefit from a figure or table. One idea would be to list/compare the proteins that are altered in EVs by stress.
Minor edits:
Line 11 mechanism of action
Line 54 avoid using possesive’s for proteins. Ie. HSP’s alternatively say “HSP proteins”
The terms exosomes and extracellular vesicles are interchangeably used, perhaps stick to extracellular vesicles since this is a broader term and also nomenclature that is preferred in the field.
Line 101 pg. 3 “cells” also in vitro and in vivo should be in italics
Line 160 pg 4 cell lines
Line 168 pg. 4 O2-2 should be in subscript
Line 175 was actual release of EVs quantified? Concentration is a balance of secretion and internalization.
Line 187 was should be were
Line 214 pg. 5 vesicle’s could be alternatively written “the proteome cargo of the vesicle”
Author Response
Response to Reviewer #3
Q1. In addition, although not necessary, the mini-review would benefit from a figure or table. One idea would be to list/compare the proteins that are altered in EVs by stress.
A1. Thank you for this suggestion. Revised manuscript has been supplemented with a table (Table 1) that summarizes relevant reports and figure (Figure 1) that illustrates pathways/processes related to proteins enhanced in EVs released by stressed cell
Q2. Line 11 mechanism of action
A2. The sentence was corrected accordingly.
Q3. Line 54 avoid using possesive’s for proteins. Ie. HSP’s alternatively say “HSP proteins”
A3. The sentence was corrected accordingly.
Q4. The terms exosomes and extracellular vesicles are interchangeably used, perhaps stick to extracellular vesicles since this is a broader term and also nomenclature that is preferred in the field.
A4. We have clarified the terminology issue in the first paragraph of the revised manuscript. Hence, term “exosome” has been replaced with term “small extracellular vesicle” or “EV” in the revised manuscript.
Q5. Line 101 pg. 3 “cells” also in vitro and in vivo should be in italics
A5. The sentence was corrected accordingly.
Q6. Line 160 pg 4 cell lines
A6. The sentence was corrected.
Q7. Line 168 pg. 4 O2-2 should be in subscript
A7. The sentence was corrected accordingly.
Q8. Line 175 was actual release of EVs quantified? Concentration is a balance of secretion and internalization.
A8. In the quoted paper only relative amounts of vesicles presented in cell culture media under different conditions were compared. Hence, the sentence was corrected: “…increased level of EVs in culture media…”.
Q9. Line 187 was should be were
A9. The sentence was corrected accordingly.
Q10. Line 214 pg. 5 vesicle’s could be alternatively written “the proteome cargo of the vesicle”
A10. The sentence has been rewritten.
Reviewer 4 Report
This review submitted by Abramowicz, Widlak and Pietrowska discusses a very important aspect related to the EV field. Though currently, EV field is on the top of the research topics, we are missing important aspect about these biological structures, mostly related to their content, their susceptibility to different stimulus and the implication of the different stress in the biology of the vesicles and their functions/applications. The review provides the description of the effect of several cellular stresses in the proteomic content of vesicles and provides frequent and appropriate citations.
Minor comments and considerations:
- In general, grammatical review and editing, publication can be recommended.
- In the abstract, line 18 “…that stress conditions affect /would affect the composition…”
- Line 29: change “…of researchers is taken in the smallest class…” by “…focused on the smallest class of vesicles…”
- Line 30: Exosomes are the vesicles comprising a sized between 30 and 100 nm, if only considering 100 nm vesicles, you are missing the smallest ones.
- Line 37: A proteomics-based approach… Which approach? It would be better to generalize and consider “proteomic approaches / proteome-based studies /The study of the vesicular proteome”
- Line 39: “…about the exosomal proteome / about the proteome of exosome” In singular.
- Line 40: Eliminate word conditions. “…to different form of stress”.
- Line 43: Eliminate word “conserved”.
- Line 45-46: re-phrase
- Overall, there are too long sentences that sometimes are difficult to follow. Be more concise.
- When talking about Exocarta and EV-proteome description, the consideration of others EV-database as EVpedia or Vesiclepedia would improve the credit of the manuscript.
- On the other hand, an only small increase in the number of heat shock proteins (HSPB1, HSPA8, HSPA1, and HSPC1) was observed in exosomes released by B-lymphoblastoid cells subjected to hyperthermia [14]. Hyperthermia increased the number of heat shock proteins (HSPA1 and HSPD1) in exosomes from the ascitic fluid of gastric cancer patients. Such exosomes promoted the maturation of dendritic cells and increased anti-tumor activity of cytotoxic T lymphocytes [15]. Not necessary to change but to consider: It may be interesting to discuss the proteomic MS/MS methodology applied in these studies or the identification of EV-marker proteins in the conditions studied (hyperthermia and controls)…as the different number of proteins identified could be different due to the methodology conditions if a shot-gun approach is applied.
- This observation was consistent with reports that the activity of exosomes released by irradiated cells promotes survival. Re-consider the sentence and clarify if the mentioned reports are referred to the following described works.
- Proteomic data presented by the same group indicated that 96 exosomes released by irradiated cells may also promote migration of recipient cells. REFERENCE?? The same as the previous study mentioned?
- Line 98: compared to non-irradiated cells? Or to EV-proteome database?
- Line 102: “Proteomic profiling revealed that among approximately 1000 proteins detected in THE analyzed EVs, there were approximately 300 proteins specific for exosomes released by irradiated cells” Rephrase “… there were approximately 300 exosome specific proteins only present after irradiation / there were around 300 proteins only present in irradiated exosomes”, for example.
- Line 118: …that among THE phosphoproteins
- Line 144: … of 48 UNIQUE proteins.
- Line 187-191: Too long sentence. Divide in two sentences.
- Line 192: “However, proteomic studies performed on exosomes derived from hypoxia-stressed (1%O2) prostate cancer cells did not reveal enrichment of extracellular matrix proteins.” REFERENCE?
- The final sentence of the paragraph does not summarize what is exposed along the Hypoxic stress section (epithelial adherence junction? Endocytosis??) Only mitochondrial damage and extracellular component, in general, is mentioned.
- Line 199-200: Recent studies indicate that glucose-related disorders may also be related to extracellular vesicles. Related to EVs release? Related to EVs functions? As it is written seems than EVs are main effectors of these disorders, or that these disorders are the cause of EV release.
- Taking into account the very different EV-isolation techniques and protocols available in the field, I would suggest to include a table summarizing the most relevant studies mentioned in the review, including the methodology used for EV-isolation, the proteomic approach applied and the most relevant finding in each study. A table is always useful in a review with so much data, and most importantly when the EV-population and the final preparation studied can vary depending on the isolation technique. Many of the referred studies also mentioned a change in EV and exosome number in the different conditions studied, including the NTA or quantification methodology in the suggested table would improve the understanding of the mentioned results.
- Review the page format along the list of references. Attention to reference page-number: 11, 24, 29, or 43 among them.
- Conclusion paragraph: line 215: proteome cargo is redundant. Consider “protein/proteomic cargo” or only “proteome”.
- Final sentence: again, which proteomic approach? I may change for “Or the application of proteomic approaches has already proven their accuracy…”
Author Response
Response to Reviewer #4
Q1. In general, grammatical review and editing, publication can be recommended.
A1. Grammatical review and editing of the revised manuscript has been performed.
Q2. In the abstract, line 18 “…that stress conditions affect /would affect the composition…”
A2. The sentence was corrected accordingly.
Q3. Line 29: change “…of researchers is taken in the smallest class…” by “…focused on the smallest class of vesicles…”
A3. The sentence was changed accordingly.
Q4. Line 30: Exosomes are the vesicles comprising a sized between 30 and 100 nm, if only considering 100 nm vesicles, you are missing the smallest ones.
A4. This was not our intention to depreciate smaller vesicles. We have rewritten this paragraph based on very recent ISEV recommendations.
Q5. Line 37: A proteomics-based approach… Which approach? It would be better to generalize and consider “proteomic approaches / proteome-based studies /The study of the vesicular proteome”
A5. The sentence was rewritten accordingly.
Q6. Line 39: “…about the exosomal proteome / about the proteome of exosome” In singular.
A6. The sentence was rewritten accordingly.
Q7. Line 40: Eliminate word conditions. “…to different form of stress”.
A7. The sentence was rewritten accordingly.
Q8. Line 43: Eliminate word “conserved”. Line 45-46: re-phrase.
A8. The sentence was corrected accordingly.
Q9. When talking about Exocarta and EV-proteome description, the consideration of others EV-database as EVpedia or Vesiclepedia would improve the credit of the manuscript.
A9. Information on other databases has been added to the revised manuscript.
Q10. On the other hand, an only small increase in the number of heat shock proteins (HSPB1, HSPA8, HSPA1, and HSPC1) was observed in exosomes released by B-lymphoblastoid cells subjected to hyperthermia [14]. Hyperthermia increased the number of heat shock proteins (HSPA1 and HSPD1) in exosomes from the ascitic fluid of gastric cancer patients. Such exosomes promoted the maturation of dendritic cells and increased anti-tumor activity of cytotoxic T lymphocytes [15]. Not necessary to change but to consider: It may be interesting to discuss the proteomic MS/MS methodology applied in these studies or the identification of EV-marker proteins in the conditions studied (hyperthermia and controls)…as the different number of proteins identified could be different due to the methodology conditions if a shot-gun approach is applied.
A10. As a matter of fact there is no proteomic study on composition of exosomes released by heat-shocked cells. Only selected HSP were analyzed in available reports. Hence, we cannot speculate on details of proteomics approaches in this context.
Q11. This observation was consistent with reports that the activity of exosomes released by irradiated cells promotes survival. Re-consider the sentence and clarify if the mentioned reports are referred to the following described works.
A11. Relevant paragraph was modified for better clarity.
Q12. Proteomic data presented by the same group indicated that 96 exosomes released by irradiated cells may also promote migration of recipient cells. REFERENCE?? The same as the previous study mentioned?
A12. Missing reference was added.
Q13. Line 98: compared to non-irradiated cells? Or to EV-proteome database?
A13. The proteomes of EVs released from irradiated and non-irradiated cells were compared; the sentence was corrected.
Q14. Line 102: “Proteomic profiling revealed that among approximately 1000 proteins detected in THE analyzed EVs, there were approximately 300 proteins specific for exosomes released by irradiated cells” Rephrase “… there were approximately 300 exosome specific proteins only present after irradiation / there were around 300 proteins only present in irradiated exosomes”, for example.
A14. The sentence was rewritten accordingly.
Q15. Line 118: …that among THE phosphoproteins.
A15. The sentence was corrected accordingly.
Q16. Line 144: … of 48 UNIQUE proteins.
A16. The sentence was corrected accordingly.
Q.17. Line 187-191: Too long sentence. Divide in two sentences.
A17. The text was modified accordingly.
Q18. Line 192: “However, proteomic studies performed on exosomes derived from hypoxia-stressed (1%O2) prostate cancer cells did not reveal enrichment of extracellular matrix proteins.” REFERENCE?
A18. Missing reference was added.
Q19. The final sentence of the paragraph does not summarize what is exposed along the Hypoxic stress section (epithelial adherence junction? Endocytosis??) Only mitochondrial damage and extracellular component, in general, is mentioned.
A19. The sentence was rewritten. It refers to data obtained in studies of prostate cancer cells (reference was added).
Q20. Line 199-200: Recent studies indicate that glucose-related disorders may also be related to extracellular vesicles. Related to EVs release? Related to EVs functions? As it is written seems than EVs are main effectors of these disorders, or that these disorders are the cause of EV release.
A20. The sentence was rewritten.
Q21. Taking into account the very different EV-isolation techniques and protocols available in the field, I would suggest to include a table summarizing the most relevant studies mentioned in the review, including the methodology used for EV-isolation, the proteomic approach applied and the most relevant finding in each study. A table is always useful in a review with so much data, and most importantly when the EV-population and the final preparation studied can vary depending on the isolation technique. Many of the referred studies also mentioned a change in EV and exosome number in the different conditions studied, including the NTA or quantification methodology in the suggested table would improve the understanding of the mentioned results.
A21. A new table (Table 1) was added to the revised manuscript. The table summarizes relevant studies quoted in this mini-review. Experimental models, methods of EVs’ purification and characterization, and proteomics approaches used in these studies are reviewed there.
Q22. Review the page format along the list of references. Attention to reference page-number: 11, 24, 29, or 43 among them.
A22. Format of references was corrected according to the journal’s standards.
Q23. Conclusion paragraph: line 215: proteome cargo is redundant. Consider “protein/proteomic cargo” or only “proteome”.
A23. The sentence was rewritten accordingly.
Q24. Final sentence: again, which proteomic approach? I may change for “Or the application of proteomic approaches has already proven their accuracy…”
A24. The sentence was rewritten accordingly.
Round 2
Reviewer 2 Report
The paper by Abramowicz et al has consistently improved its critical points. It has fixed the insufficient description of the current limits that hampers the investigation on EVs at the beginning of the text and it has provided a new useful and exhaustive Table that allow readers to understand the main features of the studies that have been compared.